# OpenReview forum: "LAMP: Data-Efficient Linear Affine Weight-Space Models for Parameter-Controlled 3D Shape Generation and Extrapolation"
_ICML.cc/2026/Conference — ICML 2026 regular_

### Official Review · Reviewer_7gSN · 2026-03-11

**Soundness:** 3
**Presentation:** 3
**Significance:** 3
**Originality:** 3
**Overall Recommendation:** 4
**Confidence:** 3

**Summary:**

This work proposes a framework for generating new 3D shapes given a set of physical parameters. The key idea is to first fit an SDF-based decoder (similar to DeepSDF) for each training shape, using a dataset of shapes with annotated physical parameters. To generate a shape with target parameters, the method creates an affine combination of decoder network weights, mixed according to the parameters of the training set. The resulting network weight is then used to decode the mesh. Experimental results on DrivAerNet++ and BlendedNet show that the framework achieves better prediction accuracy for interpolation and better generalization for extrapolation. Additionally, a safety metric is proposed to avoid invalid generation results, and an application for drag reduction demonstrates effectiveness for aerodynamic optimization.

**Compliance With Llm Reviewing Policy:**

Affirmed.

**Final Justification:**

The paper introduces a simple yet novel framework based on weight-space interpolation of SDF decoders for parameter-conditioned 3D shape generation, showing clear improvements in both interpolation and extrapolation along with a useful safety metric. While limitations such as the shared-topology assumption and the role of shared initialization were initially concerns, the rebuttal has addressed them convincingly with additional clarification and evidence.

**Key Questions For Authors:**

Nil.

**Limitations:**

yes

**Strengths And Weaknesses:**

Strength:
- To my knowledge, existing work mainly focuses on generating network weights or latents, such as HyperDiffusion. The idea of interpolating network weights with shared initialization to obtain new generation results is both interesting and novel.
- Despite its simplicity, the experimental results demonstrate that the framework achieves better performance on both DrivAerNet++ and BlendedNet datasets for interpolation and extrapolation. The framework also enables aerodynamic optimization and achieves better optimization results than the baseline.
- The proposed safety metric is shown to be effective both quantitatively and qualitatively in Figures 5 and 18–19.

Weakness:
- The current method does not naturally extend to datasets with heterogeneous topologies, which should be discussed explicitly.
- The ablation on the effectiveness of the share initialization should be included in the main experiments.

Note that I have reviewed the previous version of the paper, and the main concerns were addressed properly, so it is recommended to include those updated content in the revision.

---

> ### Author Rebuttal · Authors · 2026-03-31
>
> We thank the reviewer for the positive assessment and for recognizing the novelty, simplicity, and effectiveness of the approach, as well as the improvements over prior work and the usefulness of the safety metric.
>
> ### **Topology limitations:**
> We agree that LAMP assumes shared topology, which arises from the need for a common structural basis for weight-space alignment. This is a deliberate design choice aligned with engineering applications, where objects are typically modeled within structured parametric families (e.g., automotive or aerospace design). We will clarify this assumption explicitly and discuss it as a scope of applicability in the revised paper. This can be naturally extended by grouping shapes into topology-consistent classes and applying LAMP within each group.
>
> ---
>
> ### **Shared initialization ablation:**
> We agree this is important and will move the ablation to the main paper. Results show that shared initialization is critical: it keeps networks in a common local basin, enabling meaningful weight-space alignment and valid mixing. Without it, alignment breaks, leading to distorted geometries and loss of parameter control. (Appendix F)
>
> | **Condition**         | **R²** | **MAE** | **Mean Safe Extrapolation Range** |
> |------------------------|--------|---------|-----------------------------------|
> | **Shared Initialization**          | 0.838  | 0.507   | 330%                             |
> | **No Shared Initialization**  | -37.77 | 9.16    | 0%                               |
>
> ---
>
> We will ensure that all revised experiments, ablations, and clarified limitations are fully incorporated into the final version of the main paper.

---

> > ### Author Rebuttal · Reviewer_7gSN · 2026-04-03
> >
> > Thanks for the rebuttal. They have resolved my concerns, and I will maintain my positive rating.

---

### Official Review · Reviewer_mFs1 · 2026-03-12

**Soundness:** 2
**Presentation:** 3
**Significance:** 2
**Originality:** 3
**Overall Recommendation:** 4
**Confidence:** 4

**Summary:**

The paper proposes a weight-space based parametric 3D shape generation method. Specifically, given an object class with known parameters to control various aspects of the objects (e.g., for cars, they provided parameters such as ramp angle and trunklid length to modify those properties of a car), they find a mapping between weights and the corresponding parameters through optimization. Eventually, by manipulating these parameters they create new weights which are then decoded into a 3D shape through marching cubes. The method shows result but in interpolation and large-range extrapolation. They employ an error-checking mechanism on decoder output by comparing against linear interpolation of the combined SDFs to sanity check the output.

**Compliance With Llm Reviewing Policy:**

Affirmed.

**Final Justification:**

I move to positive rating thanks to comprehensive rebuttals addressing most of my concerns. Overall, practicality could be improved with automated mechanisms but I'm not negative anymore.

**Key Questions For Authors:**

1) Current results seems to be limited to certain type of car and airplanes. If the paper provides examples that it works on extensive range of cars or airplanes, it would be a lot more convincing.
2) Does the method requires manual work to extract parameters for each class, or is there any automated way? This is also an important criteria to evaluate the method.
3) There's a large improvement in extrapolation but not as much in interpolation, is there a reason for that?
4) How is the overall runtime/speed comparison against baselines?

**Limitations:**

yes, limitations section in appendix also a discussion in page 7

**Strengths And Weaknesses:**

Strengths:
* The paper provides interesting insight into generating parametric 3D shape models through weight-space transformation. This area is not explored that much and I believe such methods are useful to carry the weight-space methods forward.
* Paper is easy to follow and read, it doesn't require too much effort to get the idea.
* Method shows main improvements against baselines in large extrapolation.

Weaknesses:
* Soundness & Significance:
    * The paper uses aircrafts and cars as a dataset. Even though the majority of the paper shows car designs, the airplane results are not that visible in the main paper. There are some airplanes in the Figure 1. Even though they look like abstract airplanes, I believe the BlendedNet dataset provides those. To evaluate method more strongly, we should have more results in multiple categories. We also only see a specific type of car, how does it work in other vehicles (jeeps, trucks, convertibles, etc) ?
    * I'm also worried about the amount of domain knowledge or effort needed to extract the parameters (e.g., ramp angle, diffusor angle). If there's an automated way to extract these it would be very useful, but if that requires manual annotation for each class that is a bit problematic.
* Presentation
    * It felt a bit strange that an abstract has two distinct paragraphs, they usually formed as a single paragraph.

---

> ### Author Rebuttal · Authors · 2026-03-31
>
> We thank the reviewer for the constructive feedback and address concerns on dataset diversity, parameter extraction, extrapolation behavior, and computational efficiency below, and will include these additions in the final version.
>
> ### **Diversity of categories and datasets:**
> We agree that the original presentation focused more on cars. We note that **full results on the BlendedNet aircraft dataset are included in the appendix**, and in the revised version we will move representative airplane results (both interpolation and extrapolation) to the main paper for better visibility.
>
> To further strengthen diversity, we extend our evaluation to multiple **industry-level vehicle categories** beyond a single car family. Specifically, we add experiments on a sports car (Toyota Supra), an SUV (Toyota C-HR), and a convertible (Porsche Carrera GT). Using only 50 samples per category, we demonstrate accurate parametric extrapolation across these diverse designs using **direct geometric measurements (no surrogate models)**.
>
> **Table 1: Cross-vehicle parameter-controlled extrapolation and geometric fidelity across three real-world car categories (50 samples)** (repeated here for completeness, same Table 1 for 3NmE)
>
> | Method       |   |    sports car        |        |       |     SUV    |      |       |    convertible        |    |
> |---|---:|---:|---:|---:|---:|---:|---:|---:|---:|
> |          | MAE ↓ | R² ↑ | MMD ↓ (×10³) | MAE ↓ | R² ↑ | MMD ↓ (×10³) | MAE ↓ | R² ↑ | MMD ↓ (×10³) |
> | 3DShape2VecSet (cond.) | 2.76 | -3.23 | 5.50 | 3.20 | -3.57 | 3.30 | 2.80 | -2.12 | 12.00 |
> | DNI   | 1.70 | -0.06 | 0.50 | 2.89 | -0.24 | 0.74 | 3.42 | -0.37 | 0.78 |
> | AE-LPA  | 1.55 | 0.20 | 0.49 | 1.21 | 0.64 | 0.73 | 1.28 | 0.50 | 0.77 |
> | **LAMP (ours)** | 0.25 | 0.97 | 0.48 | 0.16 | 0.99 | 0.72 | 0.36 | 0.93 | 0.76 |
>
> These results demonstrate consistent generalization across distinct vehicle categories, with LAMP achieving **R² > 0.93 in all cases using only 50 samples**. This holds across different vehicle styles within structured parametric families, which is aligned with engineering design settings where topology and parametric structure are naturally consistent.
>
>
> Together with the BlendedNet aircraft results, this demonstrates that LAMP works across both **vehicle types (cars vs aircraft) and intra-class variations (sports, SUV, convertible)**.
>
> ---
>
> ### **Parameter extraction (manual vs automated):**
> The parameters used (e.g., ramp angle, diffuser angle, length, width) can be extracted automatically via simple geometric measurements from meshes (e.g., distances, angles, projections). In our experiments, these parameters are directly available in the datasets used.
>
> **In engineering settings, such parameters are typically already defined as part of the design process, making them readily accessible.** In cases where annotation is required, LAMP operates in a low-data regime (50-100 samples), keeping the annotation cost relatively small compared to large-scale generative approaches.
>
> We agree that fully automated parameter extraction is an important direction, and we will include exploring automated geometry-based parameter discovery as part of future work.
>
> ---
>
> ### **Extrapolation vs interpolation performance:**
> The smaller gains in interpolation arise because this is an in-distribution setting where performance is already near-saturated across methods, leaving limited room for improvement.
>
> In contrast, extrapolation is inherently out-of-distribution, where methods must generalize beyond observed parameter ranges without nearby examples. LAMP is specifically designed for this regime: its affine mixing formulation allows direct parameter-constrained generation beyond the dataset bounds, while preserving geometric consistency through approximate linearity in SDF weight space. As a result, it achieves substantially larger gains in extrapolation, where baseline methods tend to drift or fail to satisfy parameter constraints.
>
> ---
>
> ### **Runtime and efficiency:**
> We will add a detailed runtime comparison in the revision:
>
> - **LAMP:**
>   - Per-shape SDF overfitting: ~5 min (0.083 GPU-hours)
>   - 100 shapes: ~8.3 GPU-hours total
>   - Inference: ~5 ms
>
> - **DNI:**
>   - Same SDF cost (~8.3 GPU-hours)
>   - Additional DNI training: ~0.1 GPU-hours
>   - Inference: ~5 ms
>
> - **AE-LPA:**
>   - Training: ~10 GPU-hours
>   - Inference: ~7 s
>
> - Modern diffusion models (e.g., LION [1]):
>   - Training: ~550 GPU-hours for ~2,500 samples
>   - Inference: ~30 s
>
> Compared to the baselines, LAMP is more efficient in the low-data regime and delivers better performance in parametric control and extrapolation.
>
> [1] A. Vahdat et al., *LION: Latent Point Diffusion Models for 3D Shape Generation*, NeurIPS 2022.
>
> ---
>
> ### **Presentation:**
> We will revise the abstract to follow a single-paragraph format.

---

> > ### Author Rebuttal · Reviewer_mFs1 · 2026-04-03
> >
> > I appreciate the rebuttal and additional experiments on more car types. It provides nice insight into the paper.
> >
> > I have a few concerns on the practicality and I want to ask some questions about them.
> > If the method needs to be run on a completely new dataset of ShapeNet chairs, I believe it needs some manual work to define parameters. If I understand correctly, we need 50-100 samples and authors explained that it should be manageable to annotate. My question is wouldn't it be difficult if we want to apply the method for, say, 100 different categories?
> >
> > Again thank you very much for the effort on running the method on additional categories. My other question is somewhat related to the previous one. Did you define different parameters for each subtype (convertible, suv, sports car) and processed them independently or did you have a single parameter space and you processed all of them together? If you can process multiple categories together, I think that'd be useful.

---

> > > ### Author Response · Authors · 2026-04-04
> > >
> > > We thank the reviewer for the thoughtful questions on practicality and parameterization.
> > >
> > > **Scalability to many categories.**
> > > We agree that scaling to a very large number of unannotated categories (e.g., 100+) would require additional effort to obtain the necessary parameter annotations. This is not the main scope of the current work.
> > >
> > > Our primary goal is to address a setting that is common in engineering design: **precise, controllable shape manipulation under extremely limited data**. In these domains, annotated datasets are inherently small (often tens to hundreds of designs), and are typically already available in industry as meshes paired with parametric annotations derived from engineering drawings (e.g., vehicle platforms, airfoils, mechanical components). These parameters can also be **autonomously computed** using standard engineering pipelines, including quantities such as volume, stress/strain from simulations, or bounding box dimensions. Additionally, when annotation is required, LAMP naturally supports **partial parameterization**, allowing practitioners to annotate only the parameters of interest rather than a full parameter set. The key challenge is therefore not data collection at scale, but enabling reliable parametric control and extrapolation in this low-data regime, something existing generative models struggle with. In such settings, LAMP enables fast design-space exploration by replacing expensive geometry generation in iterative pipelines (e.g., CFD loops), reducing turnaround from days/weeks to minutes.
> > >
> > > As future work, we plan to explore this direction as a LAMP extension, enabling zero-shot control of new categories within a shape family. Concretely, this would allow editing a new car category (or chair type) in a zero-shot fashion while preserving its style, by reusing a single parametrically annotated set of ~50 cars (or chairs) and applying the same parameter space for controllable manipulation.
> > >
> > > ---
> > >
> > > **Parameter space across subtypes.**
> > > We use a **shared parameter space across all categories**. In our experiments (SUV, sports car, convertible), the same set of geometric parameters (e.g., ramp angle, diffuser angle) is defined consistently across all shapes. During inference, LAMP can simultaneously control both the **category** and the **desired parameter values**, enabling unified processing.
> > >
> > > ---
> > >
> > > Overall, we hope this clarifies that LAMP is designed for realistic low-data engineering settings while still providing a clear path toward scalable multi-category control.

---

### Official Review · Reviewer_nCUu · 2026-03-13

**Soundness:** 2
**Presentation:** 3
**Significance:** 3
**Originality:** 3
**Overall Recommendation:** 3
**Confidence:** 3

**Summary:**

This paper proposes LAMP, a low-data framework for parameter-controlled 3D shape generation. The method overfits one SDF decoder per exemplar from a shared initialization, treats the resulting weights as an aligned basis, and generates new shapes by solving for affine mixing coefficients that satisfy target parameters. The paper also proposes a linearity-mismatch metric to flag unsafe extrapolations. Experiments on DrivAerNet++ and BlendedNet cover interpolation, extrapolation, and drag-oriented optimization.

**Compliance With Llm Reviewing Policy:**

Affirmed.

**Final Justification:**

I thank the authors for their detailed rebuttal, as the new geometric measurements and compute breakdown effectively address my initial concerns regarding evaluation validity and absolute cost. However, fundamental methodological limitations remain unresolved: the method's strict reliance on shared topology restricts it to a highly specialized engineering tool, and the per-shape overfitting requirement introduces a severe scalability bottleneck. Given these inherent constraints, I decide to maintain my score.

**Key Questions For Authors:**

- What fraction of generated meshes pass the linearity-mismatch threshold in each main experiment, and are the reported tables computed before or after safety filtering? This would clarify the practical meaning of “safe extrapolation.”

- What is the actual compute cost of LAMP relative to DNI and AE-LPA?

**Limitations:**

yes

**Strengths And Weaknesses:**

**Strengths**

- The core idea is interesting. Affine mixing of aligned exemplar-specific SDF weights is simple, interpretable, and well matched to engineering-style parametric families.

- The method is conceptually clean, and the shared-initialization ablation helps support the central alignment story.

- The empirical study is fairly broad, covering two datasets, interpolation, extrapolation, optimization, and several appendix ablations.

**Weaknesses**

- My main concern is evaluation validity in the extrapolation and optimization settings. The strongest claims depend heavily on surrogate predictors trained in-distribution, while the evaluated shapes are explicitly out-of-distribution.

- The baseline set is limited for a paper making strong empirical claims. DNI and AE-LPA provide useful contrasts, but stronger conditional implicit-shape baselines or more direct parameter-conditioned alternatives would make the case more convincing.

- The scope is narrower than some of the framing implies. The method depends on shared topology, aligned local basins, and relatively structured parametric families, which are reasonable assumptions for some engineering domains but not for general 3D generation.

- The method’s data efficiency should be weighed against its optimization cost: LAMP requires overfitting one SDF network per exemplar from a shared initialization. The paper does not clearly quantify this computational burden relative to the baselines.

---

> ### Author Rebuttal · Authors · 2026-03-31
>
> We thank the reviewer for the thoughtful and constructive feedback. We address each concern below. We strengthen evaluation validity with direct geometric measurements (no surrogate), expand baselines with conditioned 3DShape2VecSet, and provide explicit compute and safety analyses. These additions reinforce the main claims and practical applicability of the method, and will be included in the revised version.
>
> ### **Evaluation validity (OOD + surrogate predictors):**
> We agree that validating extrapolation beyond surrogate predictors is important. In the current submission, we already include direct geometric evaluation for length (Appendix H), where we measure achieved geometry directly on generated meshes and observe strong agreement with surrogate-based results.
>
> To further address this concern, we add a new experiment evaluating multiple parameters directly on meshes (no surrogate) across three additional real-world vehicle categories:
> - sports car (Toyota Supra)
> - SUV (Toyota C-HR)
> - convertible (Porsche Carrera GT)
>
> We measure ramp angle, diffuser angle, length, width, roof height and car green house angle directly from geometry and observe results consistent with surrogate-based evaluation, confirming that conclusions hold in true OOD settings. This confirms that improvements are not an artifact of surrogate predictors, but reflect true geometric fidelity under OOD conditions.
>
> **Table 1: Cross-vehicle parameter-controlled extrapolation and geometric fidelity across three real-world car categories (50 samples)** (repeated here for completeness, same Table 1 for 3NmE)
>
> | Method       |   |    sports car  |     |    |     SUV    |      |       |    convertible   |    |
> |---|---:|---:|---:|---:|---:|---:|---:|---:|---:|
> |   | MAE ↓ | R² ↑ | MMD ↓ (×10³) | MAE ↓ | R² ↑ | MMD ↓ (×10³) | MAE ↓ | R² ↑ | MMD ↓ (×10³) |
> | 3DShape2VecSet (cond.) | 2.76 | -3.23 | 5.50 | 3.20 | -3.57 | 3.30 | 2.80 | -2.12 | 12.00 |
> | DNI   | 1.70 | -0.06 | 0.50 | 2.89 | -0.24 | 0.74 | 3.42 | -0.37 | 0.78 |
> | AE-LPA  | 1.55 | 0.20 | 0.49 | 1.21 | 0.64 | 0.73 | 1.28 | 0.50 | 0.77 |
> | **LAMP (ours)** | 0.25 | 0.97 | 0.48 | 0.16 | 0.99 | 0.72 | 0.36 | 0.93 | 0.76 |
>
>
> ---
>
> ### **Baselines and comparison to modern methods:**
> We selected DNI and AE-LPA because they are the only prior methods that:
>
> (i) exploit linear structure in weight/latent space, and
> (ii) operate effectively in low-data regimes (50-100 exemplars), which is the setting of LAMP.
>
> We additionally include conditioned 3DShape2VecSet, which performs poorly in this regime due to the difficulty of conditioning with limited data.
>
> More broadly, modern generative models (e.g., SDFusion, GET3D, LION) require $10^3$–$10^5$ shapes and substantial compute. In low-data regimes, they tend to collapse, memorize, or ignore conditioning signals, and do not reliably provide parametric control or extrapolation. Their output resolution is also insufficient for engineering use (e.g., LION produces only 2048-point clouds while requiring ~550 GPU-hours per class), whereas LAMP trains SDFs using 1M sampled 3D points per shape and extracts meshes at $256^3$ resolution.
>
> We further evaluated **Hunyuan3D**, a state-of-the-art mesh generator: while it achieves strong unconditional quality, mapping parameters into its latent space fails (R² ∈ [-0.1, 0.3] with 1,000 samples). This highlights that modern models lack the structured latent directions required for parametric control.
>
> ---
>
> ### **Compute cost:**
> We have added a detailed compute comparison:
>
> - **LAMP:**
>   - Per-shape SDF overfitting: ~5 min (0.083 GPU-hours)
>   - 100 shapes: ~8.3 GPU-hours total
>   - Inference: ~5 ms
>
> - **DNI:**
>   - Same SDF cost (~8.3 GPU-hours)
>   - Additional DNI training: ~0.1 GPU-hours
>   - Inference: ~5 ms
>
> - **AE-LPA:**
>   - Training: ~10 GPU-hours
>   - Inference: ~7 s
>
> - Modern diffusion models (e.g., LION [1]):
>   - Training: ~550 GPU-hours for ~2,500 samples
>   - Inference: ~30 s
>
> Compared to the baselines, LAMP is more efficient in the low-data regime and delivers better performance in parametric control and extrapolation.
>
> [1] A. Vahdat et al., *LION: Latent Point Diffusion Models for 3D Shape Generation*, NeurIPS 2022.
>
>
> ---
>
> ### **Safety filtering and validity ratio:**
> All main tables are reported before applying the linearity-mismatch safety filter.
>
> To clarify its practical effect, we will include a new table reporting the fraction of valid samples:
>
> | Experiment | % Valid (passes threshold) |
> |-----|-----|
> | Table 1    | 98% |
> | Table 2  Single Parameter  | 96% |
> | Table 2  Multi-Parameter  | 89% |
> | Table 3  Single Parameter  | 92% |
> | Table 3  Multi-Parameter  | 84% |
> | Table 4    | 97% |
>
> Additionally:
> - Appendix E studies how validity varies with sample size
> - Appendix O quantifies the extrapolation volume and shows how validity decreases smoothly with extrapolation distance
>
> These results demonstrate that LAMP enables controlled and measurable safe extrapolation.

---

> > ### Author Rebuttal · Reviewer_nCUu · 2026-04-03
> >
> > I thank the authors for the detailed rebuttal. The direct geometric measurements and compute breakdown effectively address my initial concerns regarding OOD evaluation validity and absolute cost. However, fundamental methodological issues remain:
> >
> > 1. Unaddressed Topological Constraints: The rebuttal bypasses my core concern regarding the method's strict reliance on shared topology and aligned local basins. This tacitly confirms LAMP is a highly specialized tool rather than a general 3D generation framework. The paper's framing must be revised to explicitly acknowledge these restrictive topological assumptions.
> >
> > 2. Scalability Bottleneck: The per-shape overfitting requirement means the computational cost for basis construction scales linearly ($O(N)$) with dataset size. While feasible for the evaluated micro-regime (50-100 shapes), it scales poorly to larger datasets compared to feed-forward models. This fundamental limitation must be explicitly discussed.
> >
> > Given the unaddressed topological constraints and scalability limits, I will maintain my score of 3 (Weak Reject).

---

> > > ### Author Response · Authors · 2026-04-03
> > >
> > > We thank the reviewer for the follow-up and for acknowledging that the additional experiments address the OOD evaluation and compute concerns. We address the remaining points below and will revise the framing accordingly.
> > >
> > > **Topological assumptions, scope, and significance:**
> > > We agree that LAMP is not a general-purpose 3D generation framework and should be framed more explicitly as a method for **structured, parameterized design families**. This assumption is already stated (Assumption A1: shared topology), and in practice we **construct aligned local basins** via shared initialization followed by per-shape finetuning, which enables consistent geometric directions in weight space.
> > >
> > > We will revise the paper to clearly position LAMP as targeting **low-data (50–100 samples), parameter-controlled design within structured families**, rather than general 3D generation. This design choice is driven by the problem we address: enabling precise, controllable shape manipulation under extremely limited data. In many engineering domains, curated datasets with annotated parameters are inherently small (often tens to hundreds of designs), making it impractical to train modern generative models that require thousands of samples. As a result, existing approaches cannot provide reliable parametric control in these settings. LAMP addresses this gap by enabling controllable generation and extrapolation directly in the low-data regime.
> > >
> > > Importantly, many engineering design problems are inherently constrained to shared topology families (e.g., vehicle platforms, airfoils, mechanical components), making this assumption both realistic and practically relevant. This aligns directly with real-world design pipelines, where engineers iteratively modify parameters (e.g., ramp angle, diffuser angle) and evaluate performance via expensive CFD simulations (~750 CPU hours per design in DrivAerNet++). LAMP accelerates the geometry-generation step in this loop (<10 ms solve + seconds-level meshing), enabling **rapid design exploration with performance control from days/weeks to minutes**. This setting, low-data, parameter-driven design with expensive evaluation, is common in engineering pipelines but not addressed by existing generative models.
> > >
> > > ---
> > >
> > > **Scalability and compute cost:**
> > > We agree that LAMP scales linearly with the number of shapes due to per-shape SDF overfitting, and we will explicitly discuss this limitation. Despite this linear scaling, LAMP remains practical and competitive:
> > >
> > > - **LAMP:**
> > >   - Per-shape SDF overfitting: ~5 min (0.083 GPU-hours)
> > >   - 100 shapes: ~8.3 GPU-hours
> > >   - 2,500 shapes: ~208 GPU-hours
> > >   - Inference: ~5 ms
> > >
> > > - **Diffusion models (e.g., LION):**
> > >   - Training: ~550 GPU-hours for ~2,500 samples
> > >   - Inference: ~30 s
> > >
> > > Importantly, LAMP’s cost is incurred once during basis construction, after which design exploration is effectively instantaneous, unlike diffusion models which require expensive inference for each sample.
> > >
> > > ---
> > >
> > > **Summary:**
> > > We will revise the paper to explicitly clarify the scope and assumptions (shared topology, structured families) and discuss scalability more directly. Within its intended regime, LAMP provides a unique combination of data efficiency, controllability, and extrapolation aligned with real engineering workflows.

---

### Official Review · Reviewer_3NmE · 2026-03-15

**Soundness:** 3
**Presentation:** 3
**Significance:** 2
**Originality:** 3
**Overall Recommendation:** 4
**Confidence:** 3

**Summary:**

The paper proposes a method for physically constrained optimization in latent space of decoder-only neural 3D implicit functions. The key insight of the paper is that latent space of neural implicits is locally linear and control-point map is linear as well. Authors utilize this insight to run optimization in latent space to yield shapes with target parameters. Results on DrivAerNet++ (cars) and BlendedNet (aircraft) datasets show that proposed method is on par with baselines in terms of geometry and significantly better in terms of parameter fidelity.

**Compliance With Llm Reviewing Policy:**

Affirmed.

**Final Justification:**

Authors have resolved my concerns with regard to significance of proposed work and shape artifacts and provided additional evaluation on latent grid decoders. I am raising my rating to weak accept.

**Key Questions For Authors:**

My main concern about the proposed work is its significance - it seems very niche work with very old baselines (thus weak reject rating). Would authors kindly clarify the following:

- What is the significance of proposed work beyond results on academic datasets? How transferable are results on deep SDF-like shapes to industrial settings?
- In some of the example I saw that optimized shapes have artifacts like holes in the wheels (e.g. Figure 12). How does this affect the application of the method?
- What is the reason for using decoder-only neural implicits? There are a lot of recent methods (e.g. 3DShape2VecSet, 3DILG, TRELLIS ½) with trained powerful encoders that have well structured latent spaces that might be more geometry aware.

**Limitations:**

yes

**Strengths And Weaknesses:**

**Strengths**
- Simple yet effective idea that neural implicit latent space and physical parameters space both exhibit local linearity that can be utilized for joint optimization;
- Detailed qualitative results;
- Comprehensive supplement that justifies experiments and provides additional ablations.

**Weaknesses**
- Significance of the method is not clear: application seems to be niche and baselines are 5-6 years old;
- Method relies on decoder-only approaches while more recent and more
- Quantitative improvement is very marginal with respect to chosen baselines in terms of geometric quality;

---

> ### Author Rebuttal · Authors · 2026-03-31
>
> We thank the reviewer for the thoughtful feedback and for raising important questions regarding significance, baselines, and modeling choices.
>
> ### **Significance and problem setting:**
> Existing 3D generative models are not designed for the regime required for engineering design: precise parameter control under 50-100 samples. LAMP is designed specifically for this setting. While modern generative models are designed for distribution modeling under abundant data, they do not provide **precise, interpretable control under limited data**, which is central to engineering design workflows. LAMP is specifically designed for this regime through a structured weight-space formulation.
>
> ---
>
> ### **Industrial relevance and transferability:**
> The method is directly motivated by real-world design workflows. In automotive design, engineers iteratively modify parameters (e.g., ramp angle, diffuser angle) and evaluate performance via CFD simulations, which can take ~750 CPU hours per design (DrivAerNet++).
>
> LAMP accelerates the geometry-generation step in this loop (<10 ms solve + seconds-level meshing). It does not replace downstream CFD/engineering validation, but it enables **rapid design exploration with performance control from days/weeks to minutes**. This setting, low-data, parameter-driven design with expensive evaluation, is common in engineering pipelines but not addressed by existing generative models.
>
> Importantly, DrivAerNet++ is derived from real BMW and Audi geometries. To further demonstrate transferability, we include a cross-vehicle experiment on three industrial real-world car categories: **sports car (Toyota Supra), SUV (Toyota C-HR), and convertible (Porsche Carrera GT)**.
>
> **Table 1: Cross-vehicle parameter-controlled extrapolation and geometric fidelity across three real-world car categories (50 samples)**
>
> | Method       |   |    sports car        |        |       |     SUV    |      |       |    convertible        |    |
> |---|---:|---:|---:|---:|---:|---:|---:|---:|---:|
> |          | MAE ↓ | R² ↑ | MMD ↓ (×10³) | MAE ↓ | R² ↑ | MMD ↓ (×10³) | MAE ↓ | R² ↑ | MMD ↓ (×10³) |
> | 3DShape2VecSet (cond.) | 2.76 | -3.23 | 5.50 | 3.20 | -3.57 | 3.30 | 2.80 | -2.12 | 12.00 |
> | DNI   | 1.70 | -0.06 | 0.50 | 2.89 | -0.24 | 0.74 | 3.42 | -0.37 | 0.78 |
> | AE-LPA  | 1.55 | 0.20 | 0.49 | 1.21 | 0.64 | 0.73 | 1.28 | 0.50 | 0.77 |
> | **LAMP (ours)** | 0.25 | 0.97 | 0.48 | 0.16 | 0.99 | 0.72 | 0.36 | 0.93 | 0.76 |
>
>
> ---
>
> ### **Quantitative evidence of control and geometric quality:**
> Table 1 shows that LAMP significantly improves parameter control over the strongest baseline, AE-LPA, in the low-data setting. Across all vehicle categories, LAMP achieves $R² \in [0.93, 0.99]$, compared to $[0.20, 0.64]$ for AE-LPA, indicating substantially better alignment with target parameters. This corresponds to a 4-8× reduction in MAE.
>
> Geometrically, LAMP improves **MMD by >10×** over parameter-conditioned 3DShape2VecSet (e.g., 0.0055 → 0.00048), indicating sharper and more realistic shapes. This is consistent with the representation itself: LAMP trains continuous SDFs using **1M sampled 3D points per shape** and extracts meshes at **$256^3$** resolution, whereas 3DShape2VecSet is trained with **4096 points** and operates at **$128^3$** resolution.
>
> We will include this experiment in the revised paper together with visual evidence of the improved geometric quality.
>
> ---
>
> ### **Baselines and comparison to modern generative models:**
> We compare against DNI and AE-LPA because these are the most directly comparable baselines we found for few-shot parameter-controlled generation in the **50-100 sample regime**. In contrast, modern 3D generative models typically require **$10^3-10^5$ shapes** and substantial compute, and do not provide reliable control or extrapolation in low-data settings, as also reflected by the poor conditioned 3DShape2VecSet results in Table 1.
>
> We further evaluated **Hunyuan3D** (a state-of-the-art mesh generator); while it produces strong unconditional samples, regressing parameters from its latent space yields only $R² \in [-0.1, 0.3]$ with 1,000 samples, indicating that current latent spaces are not aligned with physical parameters.
>
> We therefore adopt decoder-only SDFs because, **when overfit from a shared initialization**, they yield an aligned weight space that supports explicit affine combinations, while also providing continuous, high-resolution geometry for precise control and extrapolation.
>
> ---
>
> ### **Artifacts in wheels:**
> The observed artifacts (e.g., holes in wheels) are localized and do not affect the primary body geometry or the design parameters targeted by our method. In practical engineering workflows, wheels are typically modeled as separate components and can be replaced independently.

---

> > ### Author Rebuttal · Reviewer_3NmE · 2026-04-03
> >
> > I have carefully read authors’ replies to me and other reviewers. I don’t have additional questions for authors.

---

> > > ### Author Response · Authors · 2026-04-04
> > >
> > > We thank the reviewer for the careful reading of our rebuttal and for confirming that their concerns have been fully addressed. We truly appreciate the positive acknowledgment.
> > >
> > > Given that the concerns have been resolved and no further issues remain, we kindly invite the reviewer to reconsider the score in light of this updated assessment. We believe the clarifications and additional results strengthen the paper’s technical soundness and practical relevance.
> > >
> > > We are grateful for the reviewer’s time and thoughtful feedback.

---

### Decision · Program_Chairs · 2026-04-30

**Decision:**

Accept (regular)

**Comment:**

The paper presents LAMP, a framework for parameter-controlled 3D shape generation that operates by interpolating the weight space of SDF decoders overfit to specific exemplars. The methodology is notable for its ability to handle both interpolation and extrapolation, supported by a novel linearity-mismatch metric to identify unreliable generations. While the reviewers maintain reservations regarding the scalability and topological constraints of the approach, the consensus has shifted toward a positive recommendation following a comprehensive rebuttal.

The paper is situated at the border between a Weak Accept and a Reject. On one hand, the approach is technically sound, and the authors have been exceptionally thorough in addressing reviewer concerns regarding cost, artifacts, and comparative significance. On the other hand, the inherent limitations regarding shared topology and per-shape optimization prevent it from being a clear "Strong Accept." Given the novelty of the weight-space alignment strategy and its proven utility in drag-oriented optimization, the meta-review leans toward a Weak Accept, deferring the final judgment on the significance of these trade-offs to the Senior Area Chair (SAC).